# CacheFlow: Fast Human Motion Prediction by Cached Normalizing Flow

## Abstract

Many density estimation techniques for 3D human motion prediction require a significant amount of inference time, often exceeding the duration of the predicted time horizon. To address the need for faster density estimation for 3D human motion prediction, we introduce a novel flow-based method for human motion prediction called CacheFlow. Unlike previous conditional generative models that suffer from time efficiency, CacheFlow takes advantage of an unconditional flow-based generative model that transforms a Gaussian mixture into the density of future motions. The results of the computation of the flow-based generative model can be precomputed and cached. Then, for conditional prediction, we seek a mapping from historical trajectories to samples in the Gaussian mixture. This mapping can be done by a much more lightweight model, thus saving significant computation overhead compared to a typical conditional flow model. In such a two-stage fashion and by caching results from the slow flow model computation, we build our CacheFlow without loss of prediction accuracy and model expressiveness. This inference process is completed in approximately one millisecond, making it $4\times$ faster than previous VAE methods and $30\times$ faster than previous diffusion-based methods on standard benchmarks such as Human3.6M and AMASS datasets. Furthermore, our method demonstrates improved density estimation accuracy and comparable prediction accuracy to a SOTA method on Human3.6M. Our code and models will be publicly available.

## 1 Introduction

The task of 3D human motion prediction is to forecast the future 3D pose sequence given an observed past sequence. Traditional motion prediction methods are often based on deterministic models and can struggle to capture the inherent uncertainty in human movement. Recently, stochastic approaches have addressed this limitation. Stochastic approaches allow models to sample multiple possible future motions. Stochastic human motion prediction methods utilize conditional generative models such as generative adversarial networks (GANs) [18], variational autoencoders (VAEs) [27], and denoising diffusion probabilistic model [24]. However, many stochastic approaches cannot explicitly model the probability density distribution.

Conversely, density estimate-based approaches explicitly model the probability density distribution. In safety-critical applications such as autonomous driving [51] and human-robot interaction [29, 31, 9], a density estimate can represent all possible future motions (not just a few samples) by tracking the volume of density. It can be used to derive guarantees on safety [43, 58, 64].

However, previous density estimation suffers from high computational cost. The expensive computational cost can prohibit applications to real-time use-cases, especially with high dimensional data such as human motions. For instance, kernel density estimation (KDE) [56, 52] requires an exponentially

Submitted to 39th Conference on Neural Information Processing Systems (NeurIPS 2025). Do not distribute.

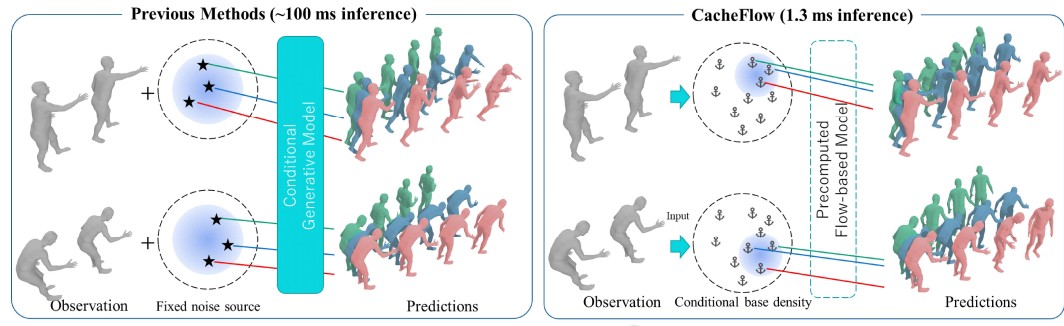

Figure 1: **Previous methods vs. Our CacheFlow.** Previous methods of stochastic motion prediction generate multiple future motions by sampling noises from the fixed source in an ad hoc manner. In contrast, CacheFlow uses the precomputed and cached latent-motion pairs from an unconditional flow-based generative model. Thus, the computation of the unconditional flow can be skipped at inference. One can achieve fast inference by selecting predictions from these cached pairs.

growing number of samples for accurate estimation. Concretely, more than one trillion samples are required for accurate KDE over a 48-dim pose over 100 frames of human motion prediction [60].

In contrast to traditional KDE, recent parametric density estimation approaches use conditional flow-based generative models, including normalizing flows [55, 62, 63] and continuous normalizing flows [13]. These flow-based generative models ("flow-based model" for brevity) directly estimate the density to avoid time-consuming sampling required in KDE. However, inferring the exact probability of possible future motions remains computationally expensive. This is because capturing the full shape of the distribution requires evaluating the probabilities of many potential future motions.

To address this computational limitation, we propose a fast density estimation method based on a flow-based model called "CacheFlow". Our CacheFlow utilizes an unconditional flow-based model for prediction, as illustrated in Figure 1. Since the unconditional flow-based model is independent of past observed motions, its calculation can be precomputed and skipped at inference. This precomputation omits a large portion of computational cost. To achieve further acceleration, our unconditional flow-based model represents transformation between a lightweight conditional base density and the density of future motions. At inference, the density of future motion is estimated by computing the lightweight conditional base density and combining it with the precomputed results of the flow-based model. The inference of our method is approximately one millisecond.

CacheFlow demonstrates comparable accuracy to previous methods on standard stochastic human motion prediction benchmarks, Human3.6M [25] and AMASS [42]. Furthermore, our method estimates density more accurately than previous stochastic human motion prediction methods with KDE. CacheFlow shows improved computational efficiency, making it well-suited for real-time applications. The contributions of this paper are four-fold as follows:

1. We introduce a novel fast density estimation called CacheFlow on human motion prediction.

2. We can sample diverse future motion trajectories with explicit density estimation, and we experimentally confirm that our method can estimate accurate density.

3. Our method achieves comparable prediction accuracy to other computationally intense methods on several benchmarks.

## 2 Related Work

### 2.1 Human Motion Prediction

**Deterministic approaches.** Early approaches on human motion prediction [1, 7, 17, 8, 21, 26, 33] focused on deterministic settings. They predict the most likely motion sequence based on the past motion. A wide range of architectures were proposed including multi-layer perceptron [21], recurrent neural networks [17, 26, 46, 20, 53, 37], convolutional neural networks [33, 50], transformers [1, 10, 48], and graph neural networks (GNNs) [44, 34, 14, 35]. GNN can account for the explicit tree

expression of the human skeleton, while other architectures implicitly learn the dependencies between joints.

**Stochastic approaches.** To capture the inherent uncertainty in human movements, recent works have focused on stochastic human motion prediction to predict multiple likely future motions. The main stream of stochastic methods use generative models for the purpose, such as generative adversarial networks (GANs) [5, 30], variational autoencoder (VAE) [65, 70, 45, 11], and denoising diffusion probabilistic model (DDPM) [4, 12, 66, 61]. To improve the diversity of predictions, diversity-promoting loss [45, 4] or explicit sampling techniques [66] were proposed. In contrast to generative models, anchor-based methods [69, 68] learn a fixed number of anchors corresponding to each prediction to ensure diversity. However, most stochastic methods cannot describe the density of future motions explicitly. This prevents exhaustive or maximum likelihood sampling for practical applications. On the contrary, our method allows for explicit density estimation using normalizing flows [28].

## 2.2 Density Estimation

Density estimation asks for explicit calculation of the probability for samples from a distribution. Density estimation is derived by non-parametric or parametric methods.

**Non-parametric Approach.** The representative non-parametric density estimation is kernel density estimation (KDE) [56, 52]. KDE can estimate density by using samples from generative models. However, KDE requires a large number of samples for accurate estimation. Therefore, it often cannot run in real-time.

**Parametric Approach.** As a representative parametric model, Gaussian mixture models (GMMs) parametrize density with several Gaussian distributions and their mixture weights. Its nature of mixing Gaussian priors limits its ability to generalize to complex data distribution. Another parametric approach with more expressivity is flow-based generative models [28]. By a learned bijective process, normalizing flows (NFs) [55, 62, 63] transform a simple density like the standard normal distribution into a complex data density. Recently, continuous normalizing flows (CNFs) [13, 19] achieve more expressive density than standard normalizing flows via an ODE-based bijective process. While training of CNFs is inefficient due to the optimization of ODE solutions, an efficient training strategy named flow matching [36] was proposed. FlowChain [40] was proposed for fast and efficient density estimation in human trajectory forecasting. FlowChain improves the inference time efficiency by reusing results from the conditional flow-based method while the past sequences are similar. However, with significantly different past sequences, FlowChain's efficiency can't hold anymore. Unlike FlowChain, our method can perform fast and efficient inference regardless of past sequences.

## 3 Preliminary

### 3.1 Problem Formulation

The task of human motion prediction aims to use a short sequence of observed human motion to predict the future unobserved motion sequence of that person. Human motion is represented by a sequence of human poses in a pre-defined skeleton format of 3D locations of $J$ joints, $X \in \mathbb{R}^{J \times 3}$. As input to our model, we have the past (history of) human motion as a sequence $\boldsymbol{c} = [X_1, X_2, ..., X_H] \in \mathbb{R}^{H \times J \times 3}$ over $H$ timesteps. To predict the future human motion sequence of $F$ timesteps, we can formulate the problem as one of conditional generation using the conditional probability function, $p(\boldsymbol{X}|\boldsymbol{c})$, where $\boldsymbol{X} = [X_{H+1}, X_{H+2}, ..., X_{H+F}] \in \mathbb{R}^{F \times J \times 3}$. Similar to the stochastic human motion prediction paradigm, the method should also allow for sampling $n$ multiple future sequences $\{\boldsymbol{X}_1, ..., \boldsymbol{X}_n\}$ from $p(\boldsymbol{X}|\boldsymbol{c})$. The focus of our work is to accelerate the inference time of estimate and sampling of the conditional density function $p(\boldsymbol{X}|\boldsymbol{c})$.

### 3.2 Normalizing Flow

Normalizing flow [55, 62, 63] is a generative model with explicit density estimation. It follows a bijective mapping $f_\theta$ with learnable parameters $\theta$. It transforms a simple base density $q(\boldsymbol{z})$ such as a Gaussian distribution into the complex data density $p(\boldsymbol{x})$. We can analytically estimate the exact

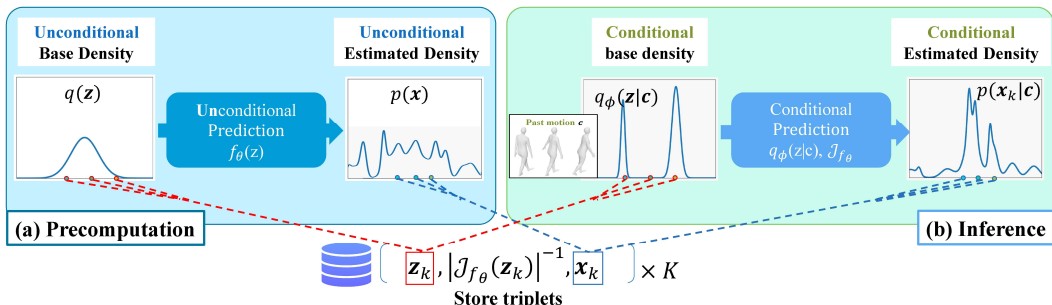

Figure 2: **Overview of our CacheFlow.** Our method utilizes the unconditional flow-based model $f_\theta$. This $f_\theta$ maps the lightweight conditional base density $q_\phi(z|c)$ into future motion density $p(x|c)$. In this formulation, the flow-based model is independent of past motions. Thus, we can precompute the unconditional flow-based model. These results are cached as K triplets as shown in (a). Due to the precomputation, we can skip the inference of $f_\theta$ and omit a large portion of the entire computation. At inference, density estimation is achieved by only evaluating the lightweight conditional base density $q_\phi(z_k|c)$ and combining it with the stored K triplets as shown in (b).

probability via the change-of-variables formula as follows:

$$x = f_\theta(z), \quad z = f_\theta^{-1}(x). \tag{1}$$

$$p(x) = q(z)|\det \mathcal{J}_{f_\theta}(z)|^{-1}, \tag{2}$$

where $\mathcal{J}_{f_\theta}(z) = \frac{\partial f_\theta}{\partial z}$ is the Jacobian of $f_\theta$ at $z$. The parameters $\theta$ of $f_\theta$ can be learned by maximizing the likelihood (or conditional likelihood) of samples $\hat{x}$ from datasets or minimizing the negative log-likelihood as $\mathcal{L}_{\text{NLL}} = -\log p(\hat{x})$. When $x$ and $z$ are latent codes, normalizing flow is transformed into latent normalizing flow. We follow this pattern in our method. We encode the past human motion into $x$ by an ecoder network $\mathcal{E}$ and decode it by a decoder network $\mathcal{D}$:

$$x = \mathcal{E}(X), X = \mathcal{D}(x). \quad x \sim \mathbb{R}^d, X \sim \mathbb{R}^{F \times J \times 3} \tag{3}$$

The encoder and decoder are trained by reconstruction. In the later part of this paper, for simplicity, we discuss the method at the latent representation level and model the conditional generation task as $p(x|c)$.

### 3.3 Continuous Normalizing Flow (CNF)

Continuous normalizing flow (CNF) [13, 19] is a normalizing flow variant based on an ordinary differential equation (ODE). CNF defines $t$-continuous path $z_t$ between the base density space $z_0 \sim q(z)$ and the data space $z_1 = x \sim p(x)$. This $z_t$ is defined by the parameterized vector field $\frac{dz_t}{dt} = v_\theta(z_t)$. The data $x = z_1$ is generated via numerical integration of vector field $v_\theta(z_t)$ as follows:

$$x = z_1 = z_0 + \int_0^1 v_\theta(z_t)dt. \tag{4}$$

The CNF transformation Equation (4) is denoted as $x = f_\theta(z)$ for brevity. Although CNF can be trained by minimizing negative log-likelihood, it is time-consuming due to the numerical integration of ODE.

### 3.4 Flow Matching

In order to train the parameterized vector field efficiently, one can leverage the flow matching [36] strategy. As a new training strategy, Flow Matching avoids the numerical integration of ODE by directly optimizing the vector field $v_\theta(z_t)$. The objective of flow matching is to match the parameterized vector field $v_\theta(z_t)$ to the ground truth vector field $u(z_t)$ via mean squared error as follows:

$$\mathcal{L}_{\text{FM}} = \mathbb{E}_{t \sim \mathcal{T}(0,1), z_t} ||v_\theta(z_t) - u(z_t)||^2, \tag{5}$$

where $\mathcal{T}(0,1)$ is a distribution ranging from 0 to 1.

However, we cannot obtain the ground truth vector field $u(\boldsymbol{z}_t)$ directly. Ripman *et al.* [36] suggest defining the conditional ground truth $u(\boldsymbol{z}_t|\hat{\boldsymbol{z}}_1)$ instead. Specifically, it is modeled as a straight vector field $\hat{\boldsymbol{z}}_1 - \boldsymbol{z}_0$ in Rectified Flow [13, 19]. This is called conditional flow matching [36] trained by the following objective:

$$\mathcal{L}_{\text{CFM}} = \mathbb{E}_{t \sim \mathcal{T}(0,1), \hat{\boldsymbol{z}}_1, \boldsymbol{z}_t} ||v_\theta(\boldsymbol{z}_t) - u(\boldsymbol{z}_t|\hat{\boldsymbol{z}}_1)||^2. \tag{6}$$

The gradients of $\mathcal{L}_{\text{FM}}$ of Equation (5) and $\mathcal{L}_{\text{CFM}}$ of Equation (6) are identical *w.r.t* $\boldsymbol{\theta}$. We exploit expressive CNFs with efficient Flow Matching training to estimate the future motion density $p(\boldsymbol{x}|\boldsymbol{c})$.

# 4 Proposed Method

## 4.1 Overview of CacheFlow

We estimate the future motion density $p(\boldsymbol{x}|\boldsymbol{c})$ by transforming a conditional base distribution $q_\phi(\boldsymbol{z}|\boldsymbol{c})$. This $q_\phi$ is conditioned on the past motion $\boldsymbol{c}$. Then we can sample predictions $\boldsymbol{x} \sim p(\boldsymbol{x}|\boldsymbol{c})$ for stochastic human motion prediction. Most traditional approaches based on conditional generative models use a trivial source distribution, often a simple Gaussian. However, we redefine the source distribution to be more informative and directly regressed from past motions. This allows us to develop a much lighter and faster model for predicting future movements.

To build this informative conditional base distribution $q_\phi$, we would incorporate an unconditional flow-based model $f_\theta : \boldsymbol{x} = f_\theta(\boldsymbol{z})$ that maps latent variable $\boldsymbol{z}$ into motion representation $\boldsymbol{x}$. To understand how $q_\phi$ and $f_\theta$ are connected, we first reparametrize the future motion density $p(\boldsymbol{x}|\boldsymbol{c})$ by a change of variables of probability equation as follows:

$$p(\boldsymbol{x}|\boldsymbol{c}) = q(\boldsymbol{z}|\boldsymbol{c}) \left| \det \frac{\partial \boldsymbol{z}}{\partial \boldsymbol{x}} \right|, \tag{7}$$

$$= q(\boldsymbol{z}|\boldsymbol{c}) \left| \det \left( \frac{\partial f_\theta(\boldsymbol{z})}{\partial \boldsymbol{z}} \right)^{-1} \right|, \tag{8}$$

$$= q(\boldsymbol{z}|\boldsymbol{c}) |\det \mathcal{J}_{f_\theta}(\boldsymbol{z})|^{-1}. \tag{9}$$

This parametrization trick differs from the widely-used conditional density formulation [67] where $\boldsymbol{c}$ is conditioned to the flow-based model $f_\theta$. In this formulation, only the conditional base density $q(\boldsymbol{z}|\boldsymbol{c})$ varies depending on $\boldsymbol{c}$ during inference, whereas the unconditional flow-based model $\boldsymbol{x} = f_\theta(\boldsymbol{z})$ and the Jacobian $|\det \mathcal{J}_{f_\theta}(\boldsymbol{z})|^{-1}$ are kept same during inference and thus can be reused as-is once calculated.

Therefore, we could precompute the mapping results and Jacobians of an unconditional flow-based model $f_\theta$. We cache the triplets $t = \{\boldsymbol{z}, |\det \mathcal{J}_{f_\theta}(z)|^{-1}, \boldsymbol{x}\}$ for later reuse in the inference stage, as shown in Figure 2(a).

Then, during inference, we design a new trick to reuse the cached triplets by associating them with the specific conditions of the past motion sequences, as shown in Figure 2(b). Now, instead of a typical conditional generative model, e.g., conditional normalizing flow, we only need a lightweight model to model the conditional base density $q_\phi(\boldsymbol{z}|\boldsymbol{c})$ and achieve similar expressivity. We could finally estimate the future motion density by $p(\boldsymbol{x}|\boldsymbol{c}) = q_\phi(\boldsymbol{z}|\boldsymbol{c})|\det \mathcal{J}_{f_\theta}(\boldsymbol{z})|^{-1}$. The method is summarized as pseudocode in Algorithm 1. In the following paragraphs, we elaborate on the details of our method.

## 4.2 Precompute Unconditional Flow-based Model

As the first step of our method, we use the human motion dataset to learn an unconditional flow-based model $f_\theta$. From this unconditional human motion prediction model, we will collect the triplets $t = \{\boldsymbol{z}, |\det \mathcal{J}_{f_\theta}(\boldsymbol{z})|^{-1}, \boldsymbol{x}\}$ for later use. This part is illustrated in Figure 2(a).

In our implementation, we built the unconditional flow model by CNFs due to its proven expressivity for predicting human motion. The unconditional model is trained to predict a fixed-length future motion $\boldsymbol{x}$ given a noise sample $\boldsymbol{z}$ from a source distribution $q_\phi(\boldsymbol{z}|\boldsymbol{c})$:

$$f_\theta : \mathbb{R}^d \longrightarrow \mathbb{R}^d \tag{10}$$

Because $z$ is sampled from a known distribution and normalizing-flow models are deterministic with reversible bijective transformation, we could know the density of each $\{z, x\}$ pair. We train the unconditional continuous normalizing flow with the flow matching objective described in Equation (6). Then we collect $K$ samples denoted by the triplet $t_k = \{z_k, |\det \mathcal{J}_{f_\theta}(z_k)|^{-1}, x_k\}$. Triplets are collected by applying the inverse transform of $f_\theta$ to ground truth future motions in the training split. These triplets are cached for fast inference as described in Section 4.3. This caching operation is different from anchor-based methods [69, 68] since CacheFlow caches all motions of the training split.

### 4.3 Conditional Inference by CacheFlow

In previous methods, conditional human motion prediction typically requires a conditional generative model. For instance, it is a conditional flow-based or diffusion model. These models usually have poor time efficiency due to delicate but heavy architecture. Instead, inspired by Equation (9), we can reuse the results of unconditional inverse transformation as triplets $t_k = \{z_k, |\det \mathcal{J}_{f_\theta}(z_k)|^{-1}, x_k\}$. Thus, we can perform conditional inference by only evaluating a conditional base distribution $q(z|c)$. We model this conditional base distribution by a learnable model, thus we denote it as $q_\phi(z|c)$. This model can be very lightweight since the unconditional transformation $f_\theta$ gives enough expressivity. $q_\phi(z|c)$ runs much faster than a typical conditional generative model for human motion prediction. This part is illustrated in Figure 2(b).

In our implementation, $q_\phi(z|c)$ is constructed as a parametrized Gaussian mixture $\{\mathcal{N}(\mu_m(c), \sigma_m^2(c))\}$, with $M$ mixture weights $w_m(c)$, such that $\sum_{m=1}^{M} w_m = 1$. Each $\mu_m$ and $\sigma_m$ are regressed based on the feature of past motion $c$. We use a lightweight single-layer RNN for regression to determine the GMM composition. Although the unconditional flow-based model $f_\theta$ and the conditional base density $q_\phi$ can be trained separately, we found that jointly training $f_\theta$ and $q_\phi$ improves model performance. We train the joint model by summation of log-likelihood for $q_\phi$ and flow matching for $f_\theta$ as explained in Equation (6) as follows:

$$\mathcal{L} = -\log q_\phi(f_\theta^{-1}(\hat{x})|c) + \mathcal{L}_{\text{CFM}}. \tag{11}$$

With joint learning, $f_\theta$ learns an easy mapping for the conditional Gaussian mixture $q_\phi$.

With $q_\phi$ constructed, during inference, we can estimate the conditional density $p(x|c)$ by connecting with precomputed triplets $t_k = \{z_k, |\det \mathcal{J}_{f_\theta}(z_k)|^{-1}, x_k\}$ as

$$p(x_k|c) = q_\phi(z_k|c)|\det \mathcal{J}_{f_\theta}(z_k)|^{-1}. \tag{12}$$

By this inference process, we could optionally generate a future human motion sequence $x$ by retrieving a high-probability sample $z$ from $q_\phi$ with the past motion sequence as the condition. However, $q_\phi$ describes a continuous distribution and the stored triplets cannot cover all samples. Therefore, in practice, predicted motion $x_{k^*}$ is selected by the nearest neighbor of the sampling outcome of $q_\phi$ to the stored triplets:

$$
\begin{aligned}
k^* &= \text{argmin}_k ||z_k - z||, \\
\text{s.t.} \quad &\{t_k = \{z_k, |\det \mathcal{J}_{f_\theta}(z_k)|^{-1}, x_k\}, \quad z \sim q_\phi(z|c)\},
\end{aligned}
\tag{13}
$$

where $k^*$ is the selected index of the triplets for prediction. By this design, we can sample an arbitrary number of likely future motion sequences by selecting the neighbors of samples $z \sim q_\phi(z|c)$.

## 5 Experimental Evaluation

**Datasets.** We evaluate our CacheFlow on Human3.6M [25] and AMASS [42]. Human3.6M contains 3.6 million frames of human motion sequences. Human motions of 11 subjects performing 15 actions are recorded at 50 Hz. We follow the setting including the dataset split, the 16-joints pose skeleton definition, and lengths of past and future motions proposed by previous works [47, 38, 71, 54]. The training and test sets of Human3.6M are subjects [S1,S5,S6,S7,S8] and [S9,S11], respectively. The past motion and future motions contain 25 frames (0.5 sec) and 100 frames (2.0 sec). AMASS unifies 24 different human motion datasets including HumanEva-I [59] with the SMPL [41] pose representation. AMASS contains 9M frames at 60 Hz in total. As a multi-dataset collection of AMASS, one can perform a cross-dataset evaluation. We follow the evaluation protocol proposed

**Algorithm 1:** Precomputation and Inference of CacheFlow.

---

**Input:** Past motion $c$
**Output:** Estimated density $p(x_k|c)$
// Precomputation. This does not count for inference time.
**for** *each future motion $X_k$ in the training dataset* **do**
   $x_k \leftarrow \mathcal{E}(X_k)$
   $z_k \leftarrow f_\theta^{-1}(x_k)$
   Calculate $|\det\mathcal{J}_{f_\theta}(z_k)|^{-1}$
   Store triplet $\{z_k, |\det\mathcal{J}_{f_\theta}(z_k)|^{-1}, x_k\}$
**end**
// Fast Inference
**for** *each triplet $\{z_k, |\det\mathcal{J}_{f_\theta}(z_k)|^{-1}, x_k\}$* **do**
   $q_\phi(z_k|c) \leftarrow \sum_{m=1}^{M} w_m\mathcal{N}(z_k; \mu_m(c), \sigma_m^2(c))$
   $p(x_k|c) \leftarrow q_\phi(z_k|c)|\det\mathcal{J}_{f_\theta}(z_k)|^{-1}$
**end**

---

| | Human3.6M [25] | | | | | AMASS [42] | | | | | Inference |
| | APD↑ | ADE↓ | FDE↓ | MMADE↓ | MMFDE↓ | APD↑ | ADE↓ | FDE↓ | MMADE↓ | MMFDE↓ | Time[ms]↓ |
|---|---|---|---|---|---|---|---|---|---|---|---|
| HP-GAN [5] | 7.214 | 0.858 | 0.867 | 0.847 | 0.858 | - | - | - | - | - | - |
| DSF [72] | 9.330 | 0.493 | 0.592 | 0.550 | 0.599 | - | - | - | - | - | - |
| DeLiGAN [23] | 6.509 | 0.483 | 0.534 | 0.520 | 0.545 | - | - | - | - | - | - |
| GMVAE [16] | 6.769 | 0.461 | 0.555 | 0.524 | 0.566 | - | - | - | - | - | - |
| TPK [65] | 6.723 | 0.461 | 0.560 | 0.522 | 0.569 | 9.283 | 0.656 | 0.675 | 0.658 | 0.674 | 30.3 |
| MT-VAE [70] | 0.403 | 0.457 | 0.595 | 0.716 | 0.883 | - | - | - | - | - | - |
| BoM [6] | 6.265 | 0.448 | 0.533 | 0.514 | 0.544 | - | - | - | - | - | - |
| DLow [73] | 11.741 | 0.425 | 0.518 | 0.495 | 0.531 | 13.170 | 0.590 | 0.612 | 0.618 | 0.617 | 30.8 |
| MultiObj [39] | 14.240 | 0.414 | 0.516 | - | - | - | - | - | - | - | - |
| GSPS [45] | 14.757 | 0.389 | 0.496 | 0.476 | 0.525 | 12.465 | 0.563 | 0.613 | 0.609 | 0.633 | 5.1 |
| Motron [57] | 7.168 | 0.375 | 0.488 | 0.509 | 0.539 | - | - | - | - | - | - |
| DivSamp [15] | 15.310 | 0.370 | 0.485 | 0.475 | 0.516 | 24.724 | 0.564 | 0.647 | 0.623 | 0.667 | 5.2 |
| BeLFusion [4] | 7.602 | 0.372 | 0.474 | 0.473 | 0.507 | 9.376 | 0.513 | 0.560 | 0.569 | 0.585 | 449.3 |
| BeLFusion-D | 5.777 | 0.367 | 0.472 | 0.469 | 0.506 | 7.458 | 0.508 | 0.567 | 0.564 | 0.591 | 39.3 |
| HumanMAC [12] | 6.301 | 0.369 | 0.480 | 0.509 | 0.545 | 9.321 | 0.511 | 0.554 | 0.593 | 0.591 | 1172.9 |
| CoMusion [61] | 7.632 | 0.350 | 0.458 | 0.494 | 0.506 | 10.848 | 0.494 | 0.547 | 0.469 | 0.466 | 352.6 |
| SLD [68] | 8.741 | 0.348 | 0.436 | 0.435 | 0.463 | - | - | - | - | - | 375.0 |
| FlowPrecomp. | 6.101 | 0.369 | 0.473 | 0.481 | 0.511 | 7.099 | 0.511 | 0.566 | 0.567 | 0.586 | 1.3 |
| w/o Precomp. | 5.385 | 0.374 | 0.489 | 0.490 | 0.531 | 6.291 | 0.516 | 0.586 | 0.573 | 0.608 | 415.9 |

Table 1: **Quantitative comparisons over the stochastic human motion prediction metrics on Human3.6M and AMASS datasets.** Lower is better for all metrics except APD. The reported inference time is when a method finishes generating 50 prediction samples from receiving the past motion.

by BeLFusion [4] for fair comparison, as predicting future 120 frames (2.0 sec) with 30 frames observation (0.5 sec) with downsampling to 60 Hz.

**Metrics.** We use the evaluation metrics to measure diversity and accuracy. 50 sampled predictions are evaluated with the following metrics: **Average Pairwise Distance (APD)** [3] evaluates sample diversity. It calculates the mean $l_2$ distance between all predicted motions. **Average and Final Displacement Error (ADE, FDE)** [2, 32, 22] evaluate accuracy. They calculate the average and final-frame $l_2$ distances between the ground truth motion and closest prediction in the 50 set. **Multimodal ADE and FDE (MMADE, MMFDE)** [72] also evaluate accuracy in a similar way to ADE and FDE. However, they are calculated over multimodal ground truths selected by grouping similar motions.

We also evaluate the accuracy of density estimation with **Multimodal Log Probability** per dimension. It calculates the log probability of the multimodal ground truths to measure how accurately the estimated density covers possible future motions. We evaluate the log probability on the motion space except for methods with latent space such as our CacheFlow and BeLFusion. While higher is better on APD and multimodal log probability, lower is better on ADE, FDE, MMADE, and MMFDE.

| Method | #sample for KDE | MM log prob. per dim ↑ | Inference Time[ms] ↓ | |
|---|---|---|---|---|
| BeLFusion | 50 | -2.383 | 2305.3 | (440.3) |
| | 1000 | -1.633 | 2422.4 | (449.3) |
| CoMusion [61] | 50 | -15.575 | 2500.5 | (167.0) |
| | 1000 | -12.746 | 5071.5 | (2741.3) |
| SLD [68] | 50 | 0.080 | 2559.1 | (375.0) |
| CacheFlow | - | **1.304** | **0.5** | (0.5) |

Table 2: **Density Estimation Accuracy on Human3.6M.** Inference time of each method is reported as {total time (time without KDE inference)}. Since our method doesn't require KDE for density estimation, the number of samples for KDE is left blank for CacheFlow.

**Implementation Details.** Our method is based on a latent flow-based model. We utilize a Variational Autoencoder (VAE) to obtain a latent representation. Specifically, we employ the Behavioral Latent Space (BLS) [4] as a VAE to achieve a compact latent representation. BLS ensures smoothness of predicted motions and consistency between the end of the past motion and the start of the predicted motion. Additionally, we compress this representation using linear factorization [68]. The dimensionality of the VAE latent space is 128, which we further reduce to 8 dimensions through linear factorization. We trained the unconditional flow-based model on this 8-dimensional space. The unconditional flow-based model $f_\theta$ is a continuous normalizing flow (CNF) model, with its vector field regressed by a U-Net architecture. The conditional base density $q_\phi$, as well as the VAE encoder and decoder, are implemented as one-layer Recurrent Neural Networks (RNNs). We used a Gaussian mixture model with $M = 50$ modes to model the conditional base density $q_\phi$. We precomputed and collected triplets $t_k = \{z_k, |\det \mathcal{J}_{f_\theta}(z_k)|^{-1}, x_k\}$ using all training samples of each dataset. All experiments, including inference time measuring, were carried out using a single NVIDIA A100 GPU. We used a batch size 64 and the Adam optimizer with a learning rate of $5 \times 10^{-4}$.

## 5.1 Quantitative Evaluation

**Accuracy Over a Fixed Number of Predictions.** We compare CacheFlow against state-of-the-art methods of stochastic human motion prediction. While we propose using a precomputed set during inference, we also evaluate our method without precomputation. In the absence of precomputation, we sample $z$ from the conditional base density $q_\phi(z|c)$ and obtain $x$ through the flow-based model inference, where $x = f_\theta(z)$. The results are summarized in Table 1. Since the primary applications of human motion prediction are in real-time scenarios, we also measure the inference time of each method to sample 50 predictions on a GPU.

CoMusion and SLD were successful in predicting motions that are closer to the ground truth than CacheFlow; however, their inference times of 167 and 375 milliseconds are too long for the intended 2000 ms prediction horizon. As a result, over 8% of the first prediction sequence is rendered useless once the prediction is finalized. Therefore, it is difficult to use these methods with slow inference in real-time applications. Although our primary goal is to estimate the density, CacheFlow achieves comparable performances with a 1.3 millisecond inference time. Our method achieves around 4× faster than the fastest VAE method, GSPS, and 30× faster than the fastest diffusion-based method, BeLFusion-D. The inference of our method is fast enough (1.3ms for future 2000ms) and applicable for real-time applications. This inference speed is because the inference of the unconditional flow-based model $f_\theta$ is precomputed. We only need to evaluate the lightweight conditional base density $q_\phi$ at inference. Although our conditional base density $q_\phi$ is just a Gaussian mixture with low expressive power, our method achieves high accuracy since the precomputed unconditional flow-based model $f_\theta$ gives $q_\phi$ much complexity with almost no overhead in inference.

**Density Estimation Accuracy.** The density estimation accuracy of each method is compared between CacheFlow and the state-of-the-art methods. The three state-of-the-art methods BeLFusion [4], CoMusion [61], and SLD [68] are selected. CoMusion and SLD were selected since they outperform our method in benchmarks of stochastic human motion prediction. We also include BeLFusion to compare CacheFlow with the method with latent space. We applied KDE to these previous methods since they only sample a set of predictions and cannot estimate density. While we evaluated 50 and 1000 samples for KDE on BeLFusion and CoMusion, SLD only allows 50 samples due to the fixed

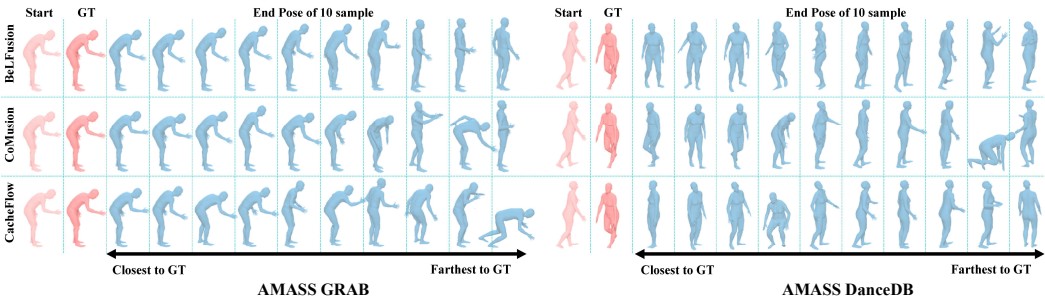

Figure 3: **Qualitative Comparison on AMASS dataset.**

number of anchors corresponding to predictions. We measured the inference time of each method to estimate the density of ten thousand future motions from the past motion input.

The quantitative comparisons over the multimodal ground truth log probability are shown in Table 2. All previous methods suffer from slow inference of their own and KDE on high-dimensional motion data. Their inference time exceeded the prediction horizon of 2000ms in the future. Therefore, they cannot estimate density in real-time. In contrast, our method achieves better estimation accuracy in less than one millisecond. This indicates that CacheFlow has strong discriminative ability to list up possible future motions required for safety assurance. Our method is even faster only on the density estimation (0.5ms) than the inference time reported in Table 1 (1.3ms). This is because we don't need any extra sampling operation in the density estimation.

## 5.2   Qualitative Comparison of Predicted Motions

To visually evaluate CacheFlow, we conducted a qualitative comparison of methods on the AMASS dataset, as shown in Figure 3. We visualized the end poses of 10 samples from each method alongside the end poses of past motions and the ground truth future motions. The sitting or lying poses were translated to the ground plane, as the global translation is not modeled in human motion prediction. The 10 pose samples are arranged from the closest to the farthest from the ground truth pose based on joint rotations.

Our observations indicate that CacheFlow predicts realistic poses. The closest poses to the ground truths also demonstrate that the accuracy of CacheFlow is comparable to CoMusion, as reflected in the ADE and FDE metrics listed in Table 1. Notably, our method is computationally efficient, operating 100 times faster than the fastest CoMusion. In summary, CacheFlow effectively delivers realistic and accurate predictions.

## 6   Concluding Remarks

We presented a new flow-based stochastic human motion prediction method named CacheFlow. Our method achieves a fast and accurate estimation of the probability density distribution of future motions. Our unconditional formulation allows precomputation and caching of the flow-based model, thus omitting a large portion of computational cost at inference. The unconditional flow-based model enhanced the expressivity of the lightweight conditional Gaussian mixture with almost no overhead. Experimental results demonstrated CacheFlow achieved comparable prediction accuracy with 1.3 milliseconds inference, much faster than the previous method. Furthermore, CacheFlow estimated a more accurate density than previous methods in less than 1 millisecond.

Our method has one limitation. Prediction and density estimation are performed within precomputed triplets. We cannot estimate the density or predict unseen future motions during precomputation. Our future work is searching for a better precomputation strategy for prediction and estimation with more coverage based on the limited dataset. Furthermore, our method is not limited to prediction tasks but applies to any regression task requiring density estimation. We will investigate the applicability of our CacheFlow on other domains.

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

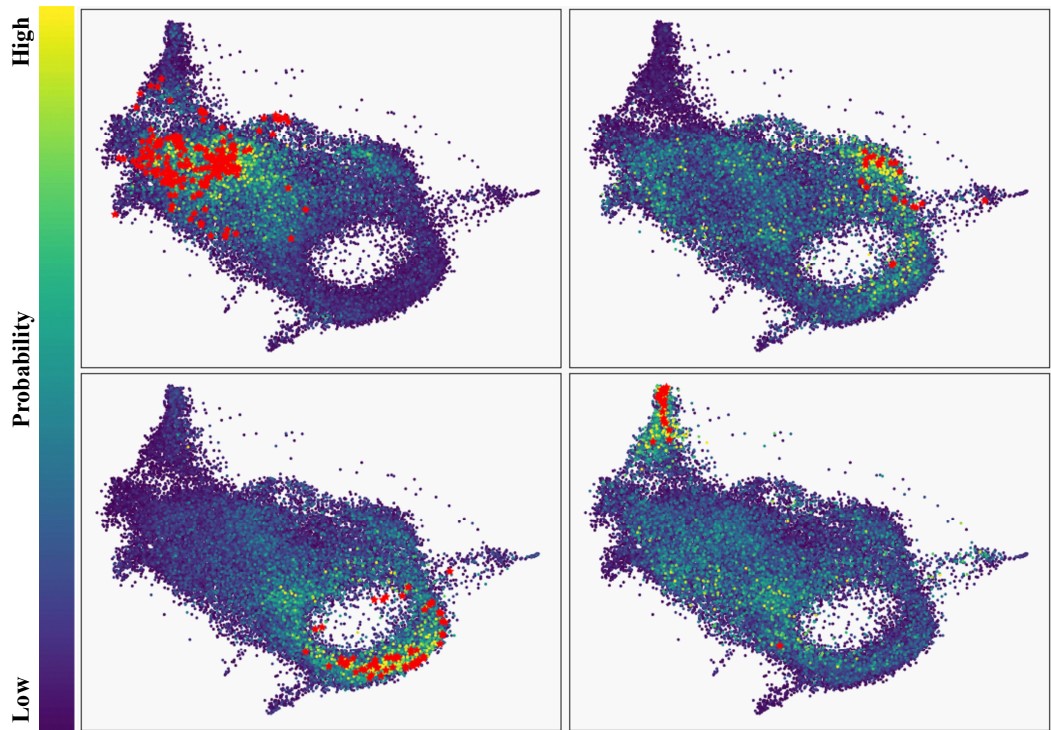

Figure 4: **Visualization of future motion densities by CacheFlow.** The estimated densities for four different motion sequences are visualized. We used UMAP to project these future motions onto a 2D space. Each dot represents an evaluated future motion, and the color of each dot indicates its probability, as shown in the side color bar. The red stars represent the projected ground truth future motions.

| | Linear Factorization | Unconditional Flow-based Model | Joint Learning | Precomp. Set | Sampling | ADE↓ | FDE↓ | MM log prob. per dims↑ | Inference time[ms]↓ |
|---|---|---|---|---|---|---|---|---|---|
| (1) | | ✓ | ✓ | Train Set | NN sample | 0.502 | 0.664 | 0.458 | 4.8 |
| (2) | ✓ | | ✓ | Train Set | NN sample | 0.616 | 0.889 | 0.901 | 0.4 |
| (3) | ✓ | ✓ | | Train Set | NN sample | 0.370 | 0.475 | 1.283 | 1.3 |
| (4) | ✓ | ✓ | ✓ | Base Density | NN sample | 0.376 | 0.492 | - | 1.3 |
| (5) | ✓ | ✓ | ✓ | Train Set | Random sample | 0.455 | 0.605 | - | **1.2** |
| | ✓ | ✓ | ✓ | Train Set | Most likely | 0.384 | 0.506 | - | 1.4 |
| | ✓ | ✓ | ✓ | Train Set | NN sample | **0.369** | **0.473** | **1.304** | 1.3 |

Table 3: **Ablation Study on Human3.6M.** (4) and (5) do not affect the ground truth log probability, these are left blank.

## A  Implementation Details of Kernel Density Estimation

We assessed the accuracy of density estimation using Kernel Density Estimation (KDE) on previous methods. To ensure a fair comparison of inference time, all KDE computations were conducted on the GPU. We applied KDE to the standardized predicted future motions (or latents for BeLFusion) to obtain the estimated density. In this process, the $i$-th dimension of the predicted future motions was standardized using its $i$-th variance, meaning that covariances were not considered during standardization. We employed Scott's rule to determine the optimal bandwidth for KDE.

## B  Ablation Study

We conducted an ablation study to investigate how each component affects the performance of our CacheFlow. We ablate five components: (1) dimensionality reduction via linear factorization on

VAE, (2) the unconditional flow-based model $f_\theta$, (3) joint learning of the conditional base density $q_\phi$ and unconditional flow-based model $f_\theta$, (4) dataset for precomputation, (5) the sampling method for metrics over a fixed number of predictions. Ablation results on the Human3.6M dataset are summarized in Table 3.

**Linear Factorization.** We first ablate the linear factorization compressing 256-dim VAE latent to be an 8-dim factor space. Our method is considerably enhanced on the compact space by avoiding the curse of dimensionality.

**The Unconditional Flow-based Model.** We ablate this flow-based model $f_\theta$ to confirm it improves the conditional base density $q_\phi$ by adding complexity. As shown in Table 3, we observe a notable performance drop without the flow-based model. Therefore, our unconditional flow-based model $f_\theta$ complements conditional base density $q_\phi$ to estimate complex density distribution over human motions.

**Joint Learning.** We ablate the joint learning of the unconditional flow-based model $f_\theta$ and the conditional base density $q_\phi$. The joint learning certainly improves both prediction errors and density estimation accuracy. The unconditional flow-based model $f_\theta$ can learn a more clustered $z$ mapped from the motion feature $x$. Thus, a conditional base density $q_\phi$ can easily model the $z$ distribution.

**Dataset for Precomputation.** We propose the precomputation over the training split. Specifically, we apply inverse transform $z = f_\theta(x)$ to ground truth future motions in the training split. However, we may precompute infinite precomputation samples. For example, we can sample $z \sim q_\phi(z|c)$ and obtain $x$ by forward transform $x = f_\theta(z)$. As shown in the ablation, precomputation on the training split outperforms one on the base density since we can regularize the prediction to be legitimate human motions using the training split.

**Sampling Method.** We propose the nearest neighbor sampling from the precomputation set as described in Section 4.3. Lastly, we ablate this sampling to evaluate its performance gain. We experimented with two sampling method alternatives: random sampling and most likely sampling. Precomputed motion features $x_{k*}$ are uniformly selected as predictions with random sampling. Most likely sampling selects motion features $x_{k*}$ with the highest probabilities $k^* = \text{argmax}_k p(x_k|c)$. We found that the large and little performance drops with random and most likely sampling respectively. This random sampling is worse due to the independence from the past motions $c$. The most likely method underperforms due to less diverse samples. It cannot select a motion feature set with diversity because all selected features are often located in one peak of the estimated density. Since ADE and FDE are best-of-many metrics, this less diversity leads to worse performance. In contrast, our sampling method is superior to others. Our sampling incorporates past motions and achieves good diversity by simulating sampling from the estimated density $p(x|c)$.

# C  Visualization of Estimated Density

We visualized the future motion density estimated by CacheFlow. Since future motions are high-dimensional data, we used UMAP [49] to project each future motion into a 2D space. We displayed the multimodal ground truth future motions alongside the visualized density map. As shown in Figure 4, CacheFlow estimated a high probability around the ground truth in all motion sequences. This visually supports the high density estimation accuracy presented in Table 2.

# D  Potential Broader Impact

The proposed CacheFlow introduces a fast probaility-aware motion prediction framework, which may involve the following broader impacts:

- **Improved Collaboration in Robotics and Automation.** In collaborative robotics and industrial automation, understanding and anticipating human motion is critical for ensuring safety and efficiency. The proposed system enables robots to predict human actions and movements with probabilistic confidence, allowing them to adjust their trajectories and tasks in real time. This leads to smoother coordination in shared workspaces such as manufacturing floors, warehouses, or hospitals, where humans and robots must work in close proximity.

- **Proactive Support in Assistive Technologies.** In assistive technologies for the elderly and individuals with disabilities, anticipating human motion is essential for delivering timely and meaningful support. A fast and uncertainty-aware human motion prediction system enables robots and smart devices to proactively assist users by foreseeing movements such as standing, walking, or reaching, even in the presence of noisy or partial sensor data. Furthermore, such a system could help prevent falls or injuries by detecting signs of instability and initiating interventions early.

- **Immersive Interactions in VR and Gaming.** Virtual reality (VR) and gaming systems stand to benefit from predictive models that can estimate future body movements in real time with associated uncertainties. This capability allows VR applications to reduce latency and create more responsive environments by anticipating user actions and gestures.

