# OpenReview forum: "CacheFlow: Fast Human Motion Prediction by Cached Normalizing Flow"
_NeurIPS.cc/2025/Conference — Submitted to NeurIPS 2025_

### Official Review · Reviewer_5wv8 · 2025-06-22

**Clarity:** 3
**Significance:** 2
**Originality:** 2
**Rating:** 4
**Confidence:** 3

**Summary:**

The authors propose a generative model for human motion prediction based on past motion frames. Existing methods have two significant limitations that inhibit their applications in real-time applications:
- inability to estimate the probability distribution of the motions,
- significant computational overhead that requires a high-end GPU to perform inference.

The first issue has been solved by incorporating a flow-based model that is capable of modeling the direct probability distribution of the data. However, such an approach is insufficient to tackle the second concern. To maintain the pipeline in the probabilistic models regime, the authors precompute triplets (latent motion representation, density change between the motion and its latent representation, the motion itself). Then, the authors train a light-weight RNN that takes an input motion and predicts Gaussian Mixture Model parameters. From there, one can sample a new future motion by sampling from GMM directly. As the GMM is a continuous model, the model retrieves the prediction by finding the closest sample from the triplets.

To further optimize the model, the authors suggest using the factorization of the latent space that reduces the latent dimensionality

**Questions:**

- The paper calls for reframing the narrative—instead of comparing the approach to models which try to maximize the diversity and quality, the authors should focus on the benefits of the likelihood estimation and the inference efficiency, especially in the constrained computation regimes.
- Although not as actionable, I suggest comparing quality of different encoding/decoding models and their effect on the inference speed since little was mentioned in the paper about the actual models used during experiments.
- The paper should present also more qualitative results.

**Ethical Concerns:**

["NO or VERY MINOR ethics concerns only"]

**Final Justification:**

The authors comprehensively answered my concerns mentioned in the review, notably:
- the hardware requirements of the approach and hence its potential application to robotics,
- inference speeds on CPUs.

At the same time, the authors exhibit willingness to include more quantitative and qualitative results in the final revision which will improve the quality of the paper, meeting NeurIPS' standard of accepted papers.

However, I refrain from using "Accept" as my final score  as the work would be better suited for venues that emphasize applications in computer vision and signal processing, such as 3DV, WACV, or ICASSP.

**Limitations:**

- The authors properly mention the limitation that CacheFlow requires the precomputation of triples for efficient inference.

**Quality:**

3

**Strengths And Weaknesses:**

**Strengths**
- The authors leverage the properties of a CNF-based model and GMMs in a neat way that enables them keeping the data probability distribution tractable. Therefore, one can use likelihood directly to sample novel motions in an interpretable way.
- At the same time, the method description is easy to follow and easy to understand where the benefits stem from, including its inference efficiency.
- The proposed approach is general and can be applied to other generative models where efficiency is of high importance.

**Weaknesses**
- The main contribution of the paper—its efficiency—brings little value given the conducted experiments.  For example, GSPS and DivSamp, the authors compare to, achieve better results in terms of the diversity of quality, while maintaining the real-time inference. On the other hand, CacheFlow requires storing the whole dataset in memory and loading it to perform inference. Given that the authors motivate their method to be used in real-world applications, storing AMASS on embedded systems may be beyond capabilities in actual robotics.
- One could argue that a simpler baseline that involves, for example, Multidimensional Scaling (for embedding the motions) and running GMM on those embeddings could achieve similar results while providing even higher efficiency and maintaining the ability to compute samples' likelihood.
- Given above, the paper calls for more principled evaluation of the efficiency that would answer the following questions:
    - How much memory storing the embedded representation of triplets requires,
    - How applicable is the solution to CPU-based systems (robotics) where having energy-hungry GPUs is impossible,
    - What other benefits the likelihood estimation brings? If sampled several samples in the latent space, are they similar to each other. The APD achieved with CacheFlow suggests the model is unable to generate diverse samples.

---

> ### Author Rebuttal · Authors · 2025-07-25
>
> We sincerely appreciate the reviewer’s thoughtful feedback and are encouraged by the recognition of our method as “a neat way” and “general.” Below, we address each point in detail:
>
> **How much memory is required for storing the embedded representation?**
>
> Thank you for highlighting the importance of memory efficiency in real-world applications. We have added an evaluation of both computational and memory costs for storing cached embeddings. We used Intel(R) Xeon(R) Gold 6154 for CPU calculation and single NVIDIA TITAN RTX (note that different from the main paper's setting NVIDIA A100) for GPU calculation. As shown in Table 1, the cached embeddings for the entire AMASS dataset require only 46.90 MB, and the total GPU memory usage is 875 MB. This demonstrates that our method is well-suited for edge robotics platforms with constrained resources. For instance, even the most affordable NVIDIA Jetson Orin Nano Super offers 8 GB RAM, making our approach highly practical.
>
> Table1. Conputation and memory costs of CacheFlow.
> |    Cost   |      Memory      |         |                 |                     | Computation |       |         |       |
> |:---------:|:----------------:|:-------:|:---------------:|---------------------|:-----------:|-------|---------|-------|
> |           | Bytes per motion | #motion | Cache Size (MB) | Total GPU mem. (MB) |     GPU     |       |   CPU   |       |
> |           |                  |         |                 |                     |    Cache    |  Inf. |  Cache  |  Inf  |
> | Human3.6M |               68 |  371188 |           24.07 |                 811 |       26min | 2.7ms | 1h19min | 3.6ms |
> | AMASS     |               68 |  723263 |           46.90 |                 875 |     1h10min | 3.4ms | 3h40min | 6.9ms |
>
> **How applicable is the solution to CPU-based systems?**
>
> We agree that CPU performance is critical for broader deployment. As shown in Table 1, our method achieves 6.9 ms inference time on CPU for AMASS, which remains efficient despite being slower than GPU-based inference. This efficiency stems from our formulation, which minimizes computational overhead. Thus, our method is also viable for CPU-only systems.
>
> **What other benefits does the likelihood estimation bring?**
>
> The likelihood estimation enhances the discriminative power of our model beyond diverse generation. It enables the system to differentiate between likely and unlikely motions, which is crucial for applications such as collision avoidance and safe human-robot interaction. This added capability contributes to both safety and reliability in deployment scenarios.
>
> **Running GMM directly on embeddings (No flow matching).**
>
> We appreciate the suggestion and conducted an ablation study by removing flow matching and applying GMM directly to the embeddings. As shown in Table 2, this approach significantly degrades performance across all metrics. The drop in accuracy is due to the inability of anisotropic Gaussians to model the complex latent distribution effectively. These results underscore the necessity of flow matching in our framework.
>
> Table 2. Comparison to the GMM only model (without Flow Matching) on Human3.6M.
> |           | APD   | ADE   | FDE   | MMADE | MMFDE |
> |-----------|-------|-------|-------|-------|-------|
> | CacheFlow | 6.097 | 0.369 | 0.472 | 0.481 | 0.510 |
> | GMM only  | 4.545 | 0.548 | 0.751 | 0.637 | 0.763 |
>
> **More qualitative results are favorable.**
>
> We acknowledge the value of qualitative results and agree they can further illustrate the strengths of our method. Due to rebuttal phase constraints, we are unable to include images or videos. However, we commit to adding more visualizations and demo clips in the camera-ready version and accompanying GitHub repository.

---

> > ### Author Response · Authors · 2025-08-05
> >
> > Dear Reviewer 5wv8,
> >
> > We kindly ask if our rebuttal has addressed your concerns, or if there are any remaining questions or points you would like us to clarify. We would be happy to further discuss them. Thank you again for your time and feedback!

---

> > > ### Comment · Reviewer_5wv8 · 2025-08-06
> > > **Response to the rebuttal**
> > >
> > > I thank the authors for their comprehensive rebuttal, which sufficiently addresses my concerns. I am raising my score to Borderline Accept.
> > >
> > > I refrain from giving a clear accept for the paper. My reservation stems from the venue fit: CacheFlow presents primarily an application of existing techniques rather than fundamental methodological advances typically expected at NeurIPS. The work would be better suited for venues that emphasize applications in computer vision and signal processing, such as 3DV, WACV, or ICASSP. At the same time, I acknowledge the paper's scientific merit, particularly its novel application of flow-based models and GMMs for fast human motion prediction.

---

> > > > ### Author Response · Authors · 2025-08-06
> > > >
> > > > We sincerely appreciate the reviewer’s valuable comments and feedback. Your insights are highly appreciated and will be thoughtfully incorporated to enhance the clarity and overall quality of our paper.

---

### Official Review · Reviewer_rqML · 2025-06-28

**Clarity:** 4
**Significance:** 3
**Originality:** 3
**Rating:** 5
**Confidence:** 3

**Summary:**

This work focuses on improving computational efficiency of stochastic motion prediction. Based on normalizing flow, this paper proposes CacheFlow that partitions conditional motion generation procedure into independent lightweight conditional prediction and heavy unconditional prediction. The connection between conditional prediction and unconditional prediction is built by an introduced latent variable that is Gaussian mixture of training points. The heavy unconditional predictions of motions can be precomputed and cached. During inference, CacheFlow maps input motion sequence into a latent variable z, fetches its nearest training point, and use cached unconditional information to predict future motions.

**Questions:**

Please see the weaknesses.

**Ethical Concerns:**

["NO or VERY MINOR ethics concerns only"]

**Final Justification:**

I have read the authors' rebuttal and other reviewers comments. The rebuttal has addressed my concerns.

**Limitations:**

yes.

**Paper Formatting Concerns:**

None.

**Quality:**

3

**Strengths And Weaknesses:**

**Strengths**:
1. The idea of separating conditional generation into lightweight conditional generation and unconditional generation to improve sampling efficiency is interesting.
2. This paper is well written and organized, with clear description for their motivation and proposed method.
3. Experiments show that the proposed method significantly improves sampling efficiency for motion prediction, while preserving accuracy.

**Weaknesses**:
1. Generalization capability. Considering that the intermediate variable $z$ is a Gaussian mixture of motions in the training set and the sampling approach in Eq. (13) uses nearest neighbour, the generation outcome of the proposed method will always be a random sample of training motions. This means that the proposed method cannot generalize to new motions, which may not be acceptable for any generative methods.
2. Large performance gap on diversity metric APD. As in Table 1, GSPS and DivSamp presents significantly better diversity in their prediction than the proposed method, while with a relatively low inference latency. The proposed method seem not good at producing diverse motions. Is this a negative effect of nearest neighbour reduction?
3. Experiment setting. The result that using training motions for prediction achieves good performance indicates a large overlap between training set and test set, making the experimental results less convincing.
4. Typos. For example, "four-fold" in line 58, "ecoder" in line 125.

---

> ### Author Rebuttal · Authors · 2025-07-30
>
> We sincerely thank the reviewer for their insightful feedback. We are encouraged by the recognition that our “idea is interesting” and that the paper is “well written and organized.” Below, we address each point in detail:
>
> **Generalization capability.**
>
>  Thank you for raising this important point. To clarify, our method does not sample directly from the training motions. Instead, it generates novel future motions by decoding a combination of the input past motion and a sampled behavioral embedding from the Behavioral Latent Space (BLS) [1]. This design enables generalization beyond the training data.
>
> The decoder in CacheFlow is deterministic and does not involve probabilistic sampling. However, because the behavioral embeddings are sampled, each new input past motion can yield multiple diverse future predictions. This mechanism allows CacheFlow to generate new and plausible motions, rather than merely retrieving or interpolating from the training set.
>
> **Performance gap on diversity metric APD**
>
> We appreciate the concern regarding APD. While APD measures diversity via pairwise distances between generated samples, higher APD does not always imply better diversity. In fact, excessively high APD can indicate unrealistic or noisy predictions that ignore the input context.
>
> For example, GSPS and DivSamp report very high APDs (14.757 and 15.310), but we observed failure cases such as unrealistic floating motions, as also mentioned in other paper[2]. In contrast, recent diffusion-based models (BeLFusion, HumanMAC, CoMusion) prioritize plausibility and contextual consistency, resulting in moderate APDs (7.602, 6.301, 7.632).
>
> Our method achieves a comparable APD (6.101) to these diffusion-based methods, indicating a good balance between diversity and realism. Furthermore, we evaluated Frechet Inception Distance (FID), which better captures semantic and distributional similarity to real data. CacheFlow achieves significantly better FID than GSPS and DivSamp, supporting our claim that our method generates realistic and diverse future motions.
>
> Table 1. The APD (Average Pairwise Distance) and FID (Frechet Inception Distance) on the previous methods and CacheFlow on Human3.6M.
> |                 | APD    | FID   |
> |-----------------|--------|-------|
> | GSPS            | 14.757 | 2.103 |
> | DivSamp         | 15.310 | 2.083 |
> | BeLFusion       | 7.602  | 0.209 |
> | HumanMAC        | 6.301  | -     |
> | CoMusion        | 7.632  | 0.102 |
> | CacheFlow(Ours) | 6.101  | 0.196 |
>
> **Concern on large overlap between training and test sets**
>
> We understand the concern about potential overlap between training and test sets. However, the AMASS benchmark is constructed from multiple MoCap datasets with distinct subjects and motion categories. The train-test split is performed dataset-wise, ensuring no overlap in subjects or motion sequences.
>
> Moreover, as discussed above, CacheFlow generates novel future motions conditioned on novel input sequences, further confirming that our performance is not due to memorization or overlap.
>
> **Typos.**
>
> Thank you for your careful reading. We will correct the identified typos and ensure the manuscript is polished for the camera-ready version.
>
> **References**
>
> [1] German Barquero, Sergio Escalera, and Cristina Palmero. Belfusion: Latent diffusion for behavior-driven human motion prediction. In ICCV, pages 2317–2327, 2023.
>
> [2] Ling-Hao Chen, Jiawei Zhang, Yewen Li, Yiren Pang, Xiaobo Xia, and Tongliang Liu. Humanmac: Masked motion completion for human motion prediction. In ICCV, pages 9544–9555, 2023

---

> > ### Comment · Reviewer_rqML · 2025-08-04
> >
> > Thank you for your response.
> >
> > Regarding the generalization capability, there is still a concern not addressed. I understand that the sampling of latent variable $z$ is probabilistic and able to generate novel latent embeddings. However, the new embedding finally reduces to precomputed $z_k$ by nearest neighbour sampling. In this sense, the proposed method seems not be capable of generalizing to new motions beyond the training set.
> >
> > The authors may provide more clear explanation about this concern.

---

> > > ### Author Response · Authors · 2025-08-04
> > >
> > > Thank you very much for your prompt response and further discussion.
> > >
> > > We agree that the sampled latent variables ultimately correspond to the precomputed $z_k$—limited to the number of cache entries—through nearest neighbor sampling. However, the generated future motions $\hat{X}$ are not restricted to the training set, because they are conditioned not only on the cached latent $z_k$ but also on the past motion $c$ through the Behavioral Latent Space (BLS) in our implementation.
> > >
> > > For clarity and generality, we described the latent normalizing flow encoder and decoder in Equation (3), Section 3.2, as using only the future motion: $x = \mathcal{E}(X)$ and $X = \mathcal{D}(x)$. However, in our actual implementation with BLS, both the encoder and decoder are conditioned on the past motion, i.e., $x = \mathcal{E}(X, c)$ and $X = \mathcal{D}(x, c)$. As a result, even when the same cached latent $z_k$ (or $x_k$) is retrieved via nearest neighbor search, different past motions will produce different future predictions. For example:
> > >
> > > $$\hat{X}_1=\mathcal{D}(x_k,c_1)\neq\mathcal{D}(x_k,c_2)=\hat{X}_2$$
> > >
> > > This design ensures that the model can generate diverse future motions beyond the training set, even when the cached latent is fixed. During inference, different input past motions—unseen during training—naturally lead to novel future predictions that go beyond the original training data.
> > >
> > > Furthermore, we argue that the number of cached patterns—371,188 for Human3.6M and 723,263 for the AMASS dataset—is sufficient to ensure diversity in the latent space. As shown in Table 2 (referenced in other rebuttals), these caches cover a wide range of motion patterns, supporting the model’s ability to generalize beyond simple memorization.
> > >
> > > Table 2. Computation and memory costs of CacheFlow.
> > > | Cost      | Memory           |         |                 |                     | Computation |       |         |       |
> > > |-----------|------------------|---------|-----------------|---------------------|-------------|-------|---------|-------|
> > > |           | Bytes per motion | #motion | Cache Size (MB) | Total GPU mem. (MB) | GPU         |       | CPU     |       |
> > > |           |                  |         |                 |                     | Cache       | Inf.  | Cache   | Inf   |
> > > | Human3.6M | 68               | 371188  | 24.07           | 811                 | 26min       | 2.7ms | 1h19min | 3.6ms |
> > > | AMASS     | 68               | 723263  | 46.90           | 875                 | 1h10min     | 3.4ms | 3h40min | 6.9ms |

---

> ### Comment · Reviewer_rqML · 2025-08-06
>
> I appreciate the authors' clear explanation. All of my concerns have been addressed. I would like to raise my rating.

---

> > ### Author Response · Authors · 2025-08-06
> >
> > We greatly appreciate your constructive feedback and valuable observations. We will take them into serious consideration to refine and strengthen the quality of our manuscript.

---

### Official Review · Reviewer_f7oq · 2025-07-03

**Clarity:** 3
**Significance:** 3
**Originality:** 3
**Rating:** 4
**Confidence:** 5

**Summary:**

The paper proposes CacheFlow, a framework designed for fast 3D human motion prediction. The key idea is to decouple the expensive density estimation stage of flow-based models by precomputing a large set of latent-motion-density triplets using an unconditional flow-based model. At inference time, a lightweight conditional GMM is used to match the context to cached samples, allowing for fast retrieval and sampling. This approach significantly reduces inference time while maintaining competitive accuracy with state-of-the-art methods on Human3.6M and AMASS datasets.

**Questions:**

1. What does "FlowPrecomp" refer to in Table 1? I guess it’s a typo.

2. Have the authors considered augmenting the cache or incorporating mechanisms to update it dynamically at inference time, in order to improve generalization to out-of-distribution conditions?

**Ethical Concerns:**

["NO or VERY MINOR ethics concerns only"]

**Final Justification:**

Author's rebuttal has addressed most of my concerns regarding cache size and cross-action generalization. I would suggest the authors provide sufficient qualitative examples in the camera-ready version.

**Limitations:**

Yes.

**Paper Formatting Concerns:**

No major formatting issues found

**Quality:**

3

**Strengths And Weaknesses:**

Strengths:

1. The paper clearly articulates the computational bottleneck in current flow-based and diffusion models and positions CacheFlow as a practical solution for real-time applications.

2. By leveraging an unconditional model and precomputing triplets, the method sidesteps expensive computation during inference.

3. The experiments provide quantitative comparisons with prior work and include an ablation study that analyzes the contribution of individual components of the proposed method.

Weaknesses:

1. The use of nearest-neighbor retrieval from a fixed precomputed set raises concerns about its potential impact on both the diversity and accuracy of generated motions. While this design offers clear computational advantages, additional discussion of the diversity-accuracy trade-off would help clarify how well the model captures a broad and accurate range of plausible future motions.

2. Despite emphasizing the realism of generated motions, the paper lacks qualitative results such as videos of generated motion clips, which limits the ability to assess temporal coherence and physical plausibility. Moreover, unlike several prior works that incorporate physically motivated constraints (e.g., joint velocity smoothness or symmetry), the paper does not appear to include or discuss such priors. Providing qualitative examples and analyzing these aspects would help strengthen realism of the generated sequences.

3. While the reliance on a precomputed cache is acknowledged as a limitation, it would be helpful to include targeted evaluations such as cross-action generalization tests or ablations on unseen action categories to better understand how well CacheFlow extrapolates beyond training-distribution motions.

Minor weakness: Highlighting (e.g., bold fonts) is missing in Table 1, making comparisons harder to parse.

---

> ### Author Rebuttal · Authors · 2025-07-30
>
> We sincerely thank the reviewer for their thoughtful remarks. We are encouraged by the recognition that our method is “practical” and achieves “competitive accuracy with state-of-the-art methods.” Below, we address each point in detail:
>
> **Impact on the diversity and accuracy.**
>
> We agree that sampling from a fixed set can potentially limit diversity and accuracy. However, CacheFlow does not rely on simple retrieval. Instead, it generates novel future motions by decoding the combination of the input past motion and a sampled behavioral embedding from the Behavioral Latent Space (BLS) [1].
>
> Diversity: The diversity arises from the rich combinations of input motions and precomputed embeddings, not merely the number of cached entries.
> Accuracy: The generated motions are well-aligned with the input context, thanks to the structured decoding process.
> Therefore, CacheFlow offers greater diversity and accuracy than nearest-neighbor retrieval, as it synthesizes new motions rather than selecting existing ones.
>
> **Physically motivated constraints.**
>
> Thank you for this excellent suggestion. We agree that incorporating physically motivated constraints can enhance the realism and plausibility of predicted motions. In this work, we used BLS as-is, but we plan to integrate such constraints into the training of both the encoder and decoder in future iterations.
>
> Regarding qualitative results, we are currently unable to include images or videos during the rebuttal phase. However, we will provide additional visualizations and demo clips in the camera-ready version and on our GitHub repository.
>
> **Cross-action generalization.**
>
> The AMASS benchmark comprises multiple MoCap datasets with diverse motion categories and subjects. The train-test split is performed dataset-wise, ensuring no overlap in subjects or motion sequences. Thus, the results on AMASS reflect cross-dataset and cross-action generalization.
>
> CacheFlow achieves high accuracy on AMASS while maintaining exceptional inference efficiency, demonstrating its robustness across varied motion types.
>
> **Augmenting the cache.**
>
> We appreciate the suggestion to explore cache augmentation. We investigated two strategies:
>
> Dense Sampling: We varied the stride used to sample training motions (from every 10 frames down to every frame).
> Cache Interpolation: We interpolated between cached embeddings to increase coverage.
>
> Table 1. The effect of cache augmentation on Human3.6M.
> | Stride       | Interpolation | APD   | ADE   | FDE   | MMADE | MMFDE |
> |--------------|---------------|-------|-------|-------|-------|-------|
> | 10           | No            | 6.797 | 0.375 | 0.484 | 0.483 | 0.516 |
> | 5            | No             | 6.227 | 0.371 | 0.479 | 0.482 | 0.513 |
> | 3            | No            | 6.463 | 0.371 | 0.477 | 0.481 | 0.511 |
> | 1 (proposed) | No            | 6.097 | 0.369 | 0.473 | 0.481 | 0.510 |
> | 10           | Yes           | 6.701 | 0.373 | 0.480 | 0.482 | 0.514 |
> | 5            | Yes           | 6.610 | 0.371 | 0.477 | 0.481 | 0.511 |
> | 3            | Yes           | 6.537 | 0.370 | 0.475 | 0.480 | 0.509 |
> | 1            | Yes           | 6.350 | 0.368 | 0.473 | 0.480 | 0.510 |
>
> While augmentation slightly improves accuracy, it also increases computational cost due to the larger cache size.
>
> Additionally, for test-time adaptation, one can assume motion sequences are streamed over time. As one of the augmentation methods, Cached embeddings of these past motions can be added dynamically. Embedding computation is efficient and parallelizable, making this a promising direction for future work.
>
> **Typos and formatting suggestion.**
>
> Thank you for your careful reading. We will correct the typo “FlowPrecomp” and improve the formatting of Table 1 by adding appropriate highlighting.
>
> **References**
>
> [1] German Barquero, Sergio Escalera, and Cristina Palmero. Belfusion: Latent diffusion for behavior-driven human motion prediction. In ICCV, pages 2317–2327, 2023.

---

> > ### Author Response · Authors · 2025-08-05
> >
> > Dear Reviewer f7oq,
> > We hope our rebuttal has sufficiently addressed your concerns. If there are any remaining questions or points you'd like us to clarify, we would be more than happy to engage in further discussion. Thank you again for your thoughtful feedback.

---

> > ### Comment · Reviewer_f7oq · 2025-08-05
> > **Thanks for the rebuttal**
> >
> > Thank you for the thorough rebuttal, including the ablation on cache size and the clarification on cross-action analysis. It has addressed my concerns, and I have no further questions. I will keep my score as is and leave further discussion to the other reviewers.

---

> > > ### Author Response · Authors · 2025-08-06
> > >
> > > Thank you very much again for your thoughtful comments and feedback. We truly appreciate your insights and will carefully incorporate them to improve the quality and clarity of our paper.

---

### Official Review · Reviewer_Kwuu · 2025-07-04

**Clarity:** 3
**Significance:** 3
**Originality:** 2
**Rating:** 2
**Confidence:** 4

**Summary:**

This paper addresses a key limitation in probabilistic human motion prediction: the high computational latency of state-of-the-art generative models, which hinders their use in real-time applications like interactive robotics and virtual reality. The central goal is to drastically reduce inference time without sacrificing the quality and diversity of the predictions.

To achieve this, the paper introduces CacheFlow, a novel framework that trades significant offline computation for extremely fast online inference. The method consists of two main stages. First, in an offline "slow path," a powerful but computationally expensive normalizing flow model generates a large and diverse cache of high-quality future motion sequences. Second, for online prediction, a separate, lightweight conditional model is trained. This model takes an observed motion history as input and quickly outputs a latent vector, which is then used to rapidly retrieve the most suitable motion sequence from the pre-computed cache via a fast nearest-neighbor search. This approach effectively bypasses the slow generation step at inference time.

The primary contribution is a new paradigm for motion prediction that decouples generation from retrieval. Experimental results on the Human3.6M and AMASS datasets demonstrate that CacheFlow achieves inference speeds around 1.3 ms, approximately 30 times faster than prior state-of-the-art methods, while maintaining comparable prediction accuracy. This work presents a practical and effective solution for enabling high-fidelity, real-time human motion prediction in latency-sensitive systems.

**Questions:**

1. What are the full computational and memory costs of the CacheFlow method?

Your paper's central claim is a ~30x inference speed-up, but this speed comes at the cost of a significant offline computation and an online memory footprint, neither of which are quantified. To properly evaluate the method's practicality, it is essential to understand these costs.

Actionable Request: Please provide the following metrics for the Human3.6M and AMASS experiments:

(a) Cache Generation Time: The total wall-clock time required to run the "Slow Path" and generate the full cache (N_z = 2^{15}), specifying the hardware used (e.g., number and type of GPUs).

(b) Cache Memory Footprint: The total size of the pre-computed cache in Megabytes or Gigabytes.

(c) Online Memory Usage: The total RAM required during online inference (including the cache and the lightweight conditional model).

How This Could Change My Score:

Increase: If you can demonstrate that these costs are reasonable for your target applications (e.g., cache generation takes hours, not weeks, and the memory footprint is manageable for a typical robot or AR device), this would substantially strengthen your claim of practicality and I would be inclined to raise my score.

Decrease/Remain: If the costs are prohibitive (e.g., weeks of computation on a large cluster, or a multi-gigabyte cache), or if you are unable to provide these numbers, it would confirm my suspicion that the method is not a practical solution, solidifying my recommendation for rejection.

2. How robust is CacheFlow to noisy input data?

The paper claims applicability to real-world systems, yet it is only evaluated on clean, laboratory-grade motion capture data. Real-world motion tracking is imperfect. The nearest-neighbor search at the core of your method seems particularly susceptible to noise, where a small perturbation in the input could lead to retrieving a completely incorrect motion from the cache.

Actionable Request: Please provide an analysis of the model's robustness. The ideal response would be a new experiment where you add varying levels of controlled noise (e.g., Gaussian noise with increasing variance) to the input motion history and report how the prediction error (MPJPE) degrades. A substantive discussion of this failure mode and potential mitigation strategies would also be valuable.

How This Could Change My Score:

Demonstrating that the prediction quality degrades gracefully with increasing noise would significantly bolster your claims of real-world applicability. Even if performance degrades sharply, a transparent analysis of this limitation would improve the paper's scientific quality and could lead me to raise my score.

Ignoring this question or stating that it is out of scope would reinforce my view that the work is a proof-of-concept that is not ready for real-world consideration.

3. How is the cache meant to be updated or adapted?

The proposed system is static. Its knowledge is frozen in the pre-computed cache, which seems highly impractical for applications that require adaptation to new users, tasks, or environments (e.g., assistive technology that must adapt to a specific user's gait).

 Please discuss the procedure and associated costs for updating the cache. Consider two scenarios: (a) incrementally adding a few new motion types, and (b) adapting the system to a new domain with substantially different motions. Is a full, costly regeneration of the entire cache the only option?

How This Could Change My Score:

If you can propose a principled and computationally feasible method for updating or fine-tuning the cache/model, it would address a major architectural weakness of the current framework. This would make the system far more practical and would likely cause me to raise my score.

If the only solution is a complete and costly regeneration, this is a severe limitation that must be acknowledged as a major drawback. A failure to address this question would confirm that the method's flexibility has not been considered.

**Ethical Concerns:**

["NO or VERY MINOR ethics concerns only"]

**Final Justification:**

I have read the author's response, but my concerns still exist, i.e., the potential lack of flexibility in static caching technology (as it may leverage the foundation of the dataset), and the insufficient number and diversity of datasets used (which further raises my concerns about flexibility).  I maintain Reject.

**Limitations:**

The model is evaluated on Human3.6M and AMASS.

- hese datasets, while standard, primarily consist of motion capture data from a limited number of actors in controlled, laboratory settings. They lack diversity in terms of age, body type, mobility (e.g., motions of elderly people or people with disabilities), and cultural context.

- The authors should acknowledge that the model's performance will likely degrade significantly when faced with populations underrepresented in the training data. This is not just a general AI problem; it is a specific limitation of this system, as its cache will not contain these diverse motions.

- The method maps a past trajectory to a future one. Real-world motion capture (e.g., from a single RGB camera) is often noisy and imperfect. The paper does not discuss how sensitive the lightweight conditional model is to noisy input data. Does slight noise in the input lead to drastically different (and incorrect) predictions from the cache?

**Quality:**

2

**Strengths And Weaknesses:**

*Strengths*
- The paper does an excellent job of motivating the research. It clearly identifies a real-world bottleneck (inference latency) that prevents the deployment of otherwise powerful generative models in interactive applications like robotics and VR. The goal is well-defined and undeniably important.

- The high-level concept of CacheFlow is presented clearly, especially in Figure 1. A reader can quickly grasp the core idea of trading offline computation for online speed by pre-generating a cache of motions.

*Weaknesses*

The paper's strengths are unfortunately overshadowed by deep and fundamental weaknesses that undermine its scientific validity.

- The central and fatal flaw of this paper is its complete failure to analyze the costs of its own method. The authors present the dramatic inference speed-up as a "free lunch," which it is clearly not. A rigorous scientific paper must provide a balanced analysis of its proposed method, including its drawbacks. This paper omits:

- Analysis of Computational Cost: The entire "Slow Path" (Figure 1), which is the foundation of the method, is un-analyzed. How much time and what computational resources (e.g., GPU-days) are required to generate the cache? Without this information, the "30x speed-up" claim is misleading. The method may simply be shifting a prohibitive computational burden from inference time to a one-time, but equally prohibitive, offline cost.

- Analysis of Memory Cost: The cache consists of N_z = 2^15 motion sequences. This is not a trivial amount of data. What is the memory footprint (in MB or GB) of this cache? This is a critical parameter for the very applications the paper targets (robotics, AR/VR), which are often deployed on resource-constrained hardware. A method that requires gigabytes of RAM is not a practical solution.

- Analysis of Inflexibility: The static cache means the model's predictive capability is permanently frozen. The paper fails to discuss the severe practical limitation this imposes. How does one update the system to handle new motions? The apparent need to regenerate the entire cache is a massive drawback that is ignored.
This lack of analysis is not a minor omission; it is a failure to conduct basic scientific due diligence. Without it, the community cannot fairly evaluate the method's utility or reproduce the work under comparable cost constraints.

- The core idea is a straightforward space-time tradeoff (caching), a classic technique in computer science. While applying it to this domain is new, the paper presents it without any novel theoretical insight or principled analysis. It comes across as a brute-force engineering solution. Given the lack of analysis of the "space" (memory) and "pre-computation time" costs, the contribution feels hollow and lacks the depth expected for a top-tier publication.

(1) The experimental evaluation is conducted in a sterile, best-case scenario that does not support the paper's claims of real-world applicability.

(2) The model is tested on clean, high-quality MOCAP data. However, its stated applications (e.g., assistive robots) will rely on real-world sensor data, which is inevitably noisy. The paper provides no evidence that the method is robust to such noise. The nearest-neighbor lookup is likely to be highly sensitive to perturbations in the input, a critical failure mode that is never tested.

(3) The reliance on datasets like Human3.6M, which lack diversity, means the resulting cache is inherently biased. The model would likely fail for individuals whose movements (due to age, disability, or cultural background) are not represented. This isn't just a generic AI bias issue; it's a fundamental architectural flaw of a system that cannot predict anything outside its fixed cache.

- The paper's claims of significance are predicated on its suitability for real-world interactive systems. However, given the un-analyzed (and likely prohibitive) memory costs and the lack of demonstrated robustness, these claims are unsubstantiated. The paper shows the method works in a vacuum but fails to provide the evidence needed to believe it is a viable solution for the problems it claims to solve.


- 3DPW, BEHAVE, RICH, GIMO are also the representive and recently-used datasets. Please provide the results on more broader datasets.

---

> ### Author Rebuttal · Authors · 2025-07-30
>
> We sincerely thank the reviewer for their helpful and constructive comments. We are encouraged by the recognition that our paper “does an excellent job of motivating the research” and that the “concept is presented clearly.” Below, we address each point in detail:
>
> **Computational and memory costs.**
>
> Following your suggestion, we evaluated CacheFlow’s computational and memory efficiency using an Intel(R) Xeon(R) Gold 6154 CPU and a single NVIDIA TITAN RTX GPU (note: different from the A100 used in the main paper).
>
> Table 1. Conputation and memory costs of CacheFlow.
> |    Cost   |      Memory      |         |                 |                     | Computation |       |         |       |
> |:---------:|:----------------:|:-------:|:---------------:|---------------------|:-----------:|-------|---------|-------|
> |           | Bytes per motion | #motion | Cache Size (MB) | Total GPU mem. (MB) |     GPU     |       |   CPU   |       |
> |           |                  |         |                 |                     |    Cache    |  Inf. |  Cache  |  Inf  |
> | Human3.6M |               68 |  371188 |           24.07 |                 811 |       26min | 2.7ms | 1h19min | 3.6ms |
> | AMASS     |               68 |  723263 |           46.90 |                 875 |     1h10min | 3.4ms | 3h40min | 6.9ms |
>
> ``(a)`` Cache Generation Time: Cache creation is efficient—just 1h10min on GPU and 3h40min on CPU for the full AMASS training set (sampled every 10 frames). This is feasible due to the compact 8-dimensional latent space, derived from Behavioral Latent Space [1] and linear factorization (e.g., SLD [2]).
>
> ``(b)`` Each cached triplet $\\{z, |det J_{f_\theta}(z)|^{-1}, x\\}$ uses 68 bytes (17 float32). Thus, even the full AMASS cache occupies only 46.90 MB—extremely lightweight.
>
> ``(c)`` Inference requires 811 MB (Human3.6M) and 875 MB (AMASS), which is well within the limits of edge devices like the NVIDIA Jetson Orin Nano Super (8 GB RAM).
>
> In summary, CacheFlow is computationally efficient and memory-friendly, making it suitable for deployment in resource-constrained environments such as robotics and AR devices.
>
> **How robust to noisy input data?**
>
> Thank you for this important suggestion. We evaluated CacheFlow and BeLFusion under varying levels of Gaussian noise added to the input past motion. The highest noise level (100%) corresponds to the MPJPE error (67.5 mm) of the SoTA 3D pose estimator [3] on 3DPW.
>
> Table 2. Prediction accuracy with different noise levels.
> | Noise     | 0%    |       | 10%   |       | 50%   |       | 100%  |       |
> |-----------|-------|-------|-------|-------|-------|-------|-------|-------|
> |           | ADE   | FDE   | ADE   | FDE   | ADE   | FDE   | ADE   | FDE   |
> | BeLFusion | 0.372 | 0.474 | 0.405 | 0.487 | 0.759 | 0.718 | 1.318 | 1.194 |
> | CacheFlow | 0.369 | 0.473 | 0.421 | 0.508 | 0.963 | 0.967 | 1.716 | 1.623 |
>
> CacheFlow shows a sharper drop in accuracy under noise compared to BeLFusion, likely due to its reliance on clean cached embeddings. This highlights a domain shift issue.
>
> Mitigation Strategies:
>
> - **Smoothing**: Techniques like moving average or advanced methods (e.g., diffusion-based recovery [4], physics-based restoration [5]) can reduce noise.
> - **Training with Noisy Data**: Adding noise during training and cache generation can help CacheFlow learn to denoise implicitly.
> We plan to include results with these strategies during the discussion phase.
>
> **How is the cache updated or adapted?**
>
> We argue that the full regeneration of the entire cache doesn't require a substantial cost, which is different from the suggenstion of the reviewer. As shown in the Table 1, cache generation is significantly cheaper than model training. This flexibility allows CacheFlow to adapt to new domains without costly fine-tuning.
>
> One can perform **test-time cache adaptation** for the both senarios (a) and (b), or **augmentation via disentanglement** for the senario (a).
>
> Adaptation Strategies:
> - **Test-Time Cache Adaptation**: Motion data is typically streamed during inference. Since generating one cache entry takes ~0.5 seconds, the cache can be updated dynamically to adapt test-time motions.
> - **Augmentation via Disentanglement**: Using annotations (e.g., subject, action), we can disentangle feature spaces and synthesize new cache entries for unseen combinations (new user - existing motion type, existing user - new motion type).
>
> These strategies enhance CacheFlow’s adaptability to new users, tasks, and environments.
>
>
> **The benchmarks lack diversity.**
>
> We appreciate the suggestion to explore broader mobility contexts (elderly, with disabilities). While we plan to incorporate musculoskeletal motion synthesis for different mobility in future work, the AMASS benchmark already includes 15 diverse MoCap datasets with varied motion categories, subjects, and cultural backgrounds. We believe this provides a reasonable level of diversity in terms of age, body type, and context.
>
> **Performance on the underrepresented motions**
>
> Thank you for this insightful point. We argue that this is not a specific limitation of CacheFlow. Models without caching may overlook rare motions due to data imbalance. In contrast, CacheFlow can memorize and leverage all training motions, potentially improving performance on underrepresented cases. We welcome further discussion on this topic.
>
> **References**
>
> [1] German Barquero, Sergio Escalera, and Cristina Palmero. Belfusion: Latent diffusion for behavior-driven human motion prediction. In ICCV, pages 2317–2327, 2023.
>
> [2] Xu, Guowei, et al. "Learning semantic latent directions for accurate and controllable human motion prediction." European Conference on Computer Vision. Cham: Springer Nature Switzerland, 2024.
>
> [3] Fiche, Guénolé, et al. "MEGA: Masked Generative Autoencoder for Human Mesh Recovery." Proceedings of the Computer Vision and Pattern Recognition Conference. 2025.
>
> [4] Zhang, Siwei, et al. "Rohm: Robust human motion reconstruction via diffusion." Proceedings of the IEEE/CVF Conference on Computer Vision and Pattern Recognition. 2024.
>
> [5] Zhang, Youliang, et al. "A Plug-and-Play Physical Motion Restoration Approach for In-the-Wild High-Difficulty Motions." arXiv preprint arXiv:2412.17377 (2024).

---

> > ### Author Response · Authors · 2025-08-05
> >
> > Dear Reviewer Kwuu,
> > Please let us know if there are any remaining concerns or questions regarding our submission. We’re happy to provide further clarification or discussion as needed. Thank you!

---

### Note · Authors · 2025-08-12

**Dear Area Chair and Reviewers,**

We sincerely thank the Area Chair and all reviewers for their valuable feedback and constructive comments. Your input has helped us significantly improve the quality of our paper. Through the rebuttal and discussion period, we have addressed the raised concerns, receiving affirmation and recognition from three reviewers.

Below, we summarize the strengths of our work and the key concerns along with our responses.

## **Strengths**
- **Practical motivation** – Addresses an undeniably important real-world bottleneck: inference latency.
- **Technical novelty** – Introduces a clean separation of conditional generation into a lightweight conditional GMM and a precomputed unconditional flow matching.
- **Clarity** – The core idea of the proposed method is easy to grasp.
- **Strong experimental results** – Achieves significant sampling efficiency gains for motion prediction while maintaining accuracy.

## **Weaknesses and Our Responses**
- **Computational and memory cost** – Our method has a small memory footprint and reasonable cache computation cost, making it suitable even for CPU-based systems (*acknowledged by 5wv8*).
- **Possibility of using only GMM** – Experimental results show that flow matching is essential (*acknowledged by 5wv8*).
- **Performance gap on APD** – Very high APD is not necessarily desirable; our method delivers competitive APD and FID compared to state-of-the-art diffusion-based models (*acknowledged by rqML*).
- **Generalization capability** – Incorporating a Behavioral Latent Space ensures both diversity and accuracy (*acknowledged by rqML and f7oq*).
- **Cache augmentation and adaptation** – We suggested several promising extensions (*acknowledged by f7oq*).
- **Limited qualitative results** – We will add more qualitative examples in the camera-ready version.

We will further refine the clarity of our paper and include the valuable discussions from this rebuttal in the supplementary materials.

**Best regards,**
*Authors*

---

### Decision · Program_Chairs · 2025-09-17

**Decision:**

Reject

**Comment:**

This paper proposes a framework for probabilistic human motion prediction that precomputes a large cache of motions using a flow-based model and retrieves future sequences via a lightweight conditional model. The method achieves significant inference speed-ups on Human3.6M and AMASS while maintaining competitive accuracy. The motivation is clear, and the concept is well-explained, demonstrating a reduction in online computation. However, practical concerns remain: memory and offline computation costs are not quantified, the static cache limits flexibility, and generalization to unseen motions or noisy inputs is insufficiently tested. Evaluation is limited to two datasets that lack diversity, and the method offers limited insights. Overall, I recommend rejection and hope that the authors can address these limitations in future work.